# An Empirical Investigation of Representation Learning for Imitation

**Xin Chen**[*]
The University of Hong Kong
cyn0531@connect.hku.hk

**Sam Toyer**[*]
UC Berkeley
sdt@berkeley.edu

**Cody Wild**[*]
UC Berkeley
codywild@berkeley.edu

**Scott Emmons**
UC Berkeley

**Ian Fischer**
Google Research

**Kuang-Huei Lee**
Google Research

**Neel Alex**
UC Berkeley

**Steven Wang**
UC Berkeley

**Ping Luo**
The University of Hong Kong

**Stuart Russell**
UC Berkeley

**Pieter Abbeel**
UC Berkeley

**Rohin Shah**
UC Berkeley

## Abstract

Imitation learning often needs a large demonstration set in order to handle the full range of situations that an agent might find itself in during deployment. However, collecting expert demonstrations can be expensive. Recent work in vision, reinforcement learning, and NLP has shown that auxiliary representation learning objectives can reduce the need for large amounts of expensive, task-specific data. Our Empirical Investigation of Representation Learning for Imitation (EIRLI) investigates whether similar benefits apply to imitation learning. We propose a modular framework for constructing representation learning algorithms, then use our framework to evaluate the utility of representation learning for imitation across several environment suites. In the settings we evaluate, we find that existing algorithms for image-based representation learning provide limited value relative to a well-tuned baseline with image augmentations. To explain this result, we investigate differences between imitation learning and other settings where representation learning *has* provided significant benefit, such as image classification. Finally, we release a well-documented codebase which both replicates our findings and provides a modular framework for creating new representation learning algorithms out of reusable components.

## 1 Introduction

Much recent work has focused on how AI systems can learn what to do from human feedback [1]. The most popular approach—and the focus of this paper—is *imitation learning* (IL), in which an agent learns to complete a task by mimicking demonstrations of a human.

As demonstrations can be costly to collect, we would like to learn representations that lead to better imitation performance given limited data. Many existing representation learning (RepL) methods in Computer Vision and Reinforcement Learning do exactly this, by extracting effective visual [2] or temporal [3] information from inputs. A natural hypothesis is that RepL would also add value for IL.

We test this hypothesis by investigating the impact of common RepL algorithms on Behavioral Cloning (BC) and Generative Adversarial Imitation Learning (GAIL). We survey a wide variety of RepL methods, and construct a modular framework in which each design decision can be varied

---

[*]Equal contribution, corresponding authors

35th Conference on Neural Information Processing Systems (NeurIPS 2021) Track on Datasets and Benchmarks.

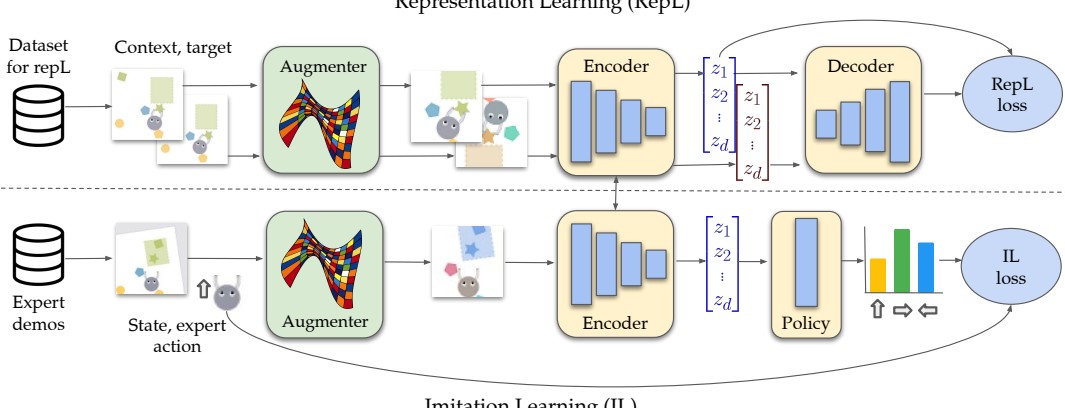

Figure 1: A framework for the use of representation learning (RepL) in imitation learning. In the pretraining setting, we first train the encoder with RepL, then finetune end-to-end with IL. In the joint training setting, the RepL objective is used as an auxiliary loss throughout IL training.

independently. As previous work has found that image augmentation alone can outperform more complex representation learning techniques [4, 5], we make sure to compare against baselines that use augmentation. To ensure generalizability of our results, we evaluate on ten tasks selected across three benchmarks, including MAGICAL [6], Procgen [7] and the DeepMind Control Suite (DMC) [8].

We find that, on average, RepL methods do significantly outperform vanilla BC, but this benefit can be obtained simply by applying well-tuned image augmentations during BC training. To understand the discrepancy between this result and the success of RepL in computer vision and reinforcement learning, we apply clustering algorithms and attribution methods to qualitatively investigate the learned representations and policies, surfacing a number of intriguing hypotheses for investigation in future work.

This paper is, to the best of our knowledge, the first to provide a systematic empirical analysis of different representation learning methods for imitation learning in image-based environments. Concretely, our Empirical Investigation of Representation Learning for Imitation (EIRLI) makes the following contributions:

1. We identify meaningful axes of variation in representation learning algorithm design, allowing us to construct a modular framework to conceptually analyze these designs.

2. We use this framework to build a well documented, modular, and extensible code base, which we release at `github.com/HumanCompatibleAI/eirli`.

3. We conduct an extensive comparison of popular RepL methods in the imitation learning setting, and show that RepL has limited impact on task performance relative to ordinary image augmentations. By analysing our learned representations and policies, we identify several promising directions for future work at the intersection of representation learning and decision-making.

## 2   Design decisions in representation learning

To apply representation learning (RepL) effectively, it is important to understand the relative impact of different RepL algorithm design choices on downstream task performance. We argue that for many common RepL algorithms, these design choices can be broken down along a common set of axes, which we show in Table 1 and Table 2. In this section, we elaborate on our conceptual breakdown both as a literature review and as an implementation walkthrough of our RepL framework.

We summarize existing RepL for image classification algorithms in Table 1 and a selection of RepL for reinforcement learning algorithms in Table 2. The full version of the table deconstructing current RepL methods in reinforcement learning can be found in the appendix in Table A1. [5]

Table 1: Design choices made in representation learning for image recognition. "Augmentation", "Momentum", and "Projection" show whether image augmentation, target encoder momentum, and projection heads were used, respectively. "Pre/Joint" shows whether RepL is used as a pretraining step, or is jointly learned with the downstream task (typically as an auxiliary loss).

| Algorithm | Task | Augmentation | Momentum | Projection | Pre/Joint |
|---|---|---|---|---|---|
| VAE [9] | Reconstruction | ✗ | ✗ | ✗ | Pre |
| AugMix [10] | Consistency | ✓ | ✗ | ✗ | Joint |
| FixMatch [11] | Consistency | ✓ | ✗ | ✗ | Joint |
| CPC [12] | Contrastive | ✗ | ✗ | ✓ | Pre |
| MoCo [13] | Contrastive | ✓ | ✓ | ✗ | Pre |
| SimCLR [2] | Contrastive | ✓ | ✗ | ✓ | Pre |
| SimCLRv2 [14] | Contrastive | ✓ | ✗ | ✓ | Pre |
| BYOL [15] | Bootstrap | ✓ | ✓ | ✓ | Pre |

Table 2: Design choices made in a selection of representation learning algorithms for reinforcement learning (full table in the appendix). Act, Aug, Mom, Proj and Comp respectively show whether action conditioning, augmentation, momentum, projection heads, and compression were used. P/J determines whether the representation learning is an initial (P)retraining step, or is (J)ointly learned alongside reinforcement learning. R/C in the Task column refer to Reconstruction/Contrastive. Note that different papers may use different sets of augmentations.

| Algorithm | Task | RL alg. | Context | Target | Act | Aug | Mom | Proj | Comp | P/J |
|---|---|---|---|---|---|---|---|---|---|---|
| World models [16] | R | CMA-ES | $o_t$ | $o_t, o_{t+1}$ | ✓ | ✗ | ✗ | ✗ | ✗ | P |
| PlaNet [17] | R | MPC + CEM | $o_{1:t}$ | $o_{t+1:T}, r_{t+1:T}$ | ✓ | ✗ | ✗ | ✗ | ✗ | J |
| CURL [18] | C | SAC | $o_t$ | $o_t$ | ✗ | ✓ | ✗ | ✗ | ✗ | J |
| PI-SAC [3] | C | SAC | $o_t$ | $o_{t+k}, r_{t+k}$ | ✓ | ✓ | ✓ | ✓ | ✓ | J |
| ATC [19] | C | SAC, PPO | $o_t$ | $o_{t+k}$ | ✗ | ✓ | ✓ | ✓ | ✗ | P |

## 2.1 Target selection

Most RepL methods can be thought of as proxy tasks in which a dataset of $(x, y)$ pairs is provided and the network must model some aspects of the relationship between $x$ and $y$. Since the learning signal derives from the relationship between $x$ and $y$, the choice of $x$ and $y$ thus has a significant impact on exactly what information is modeled. We refer to the inputs for which representations $z$ are computed as the "context" $x$, and the inputs with which they are related are the "targets" $y$. Often, the target is a (possibly transformed) version of a context.

In image classification, learned representations must capture the label-relevant information in a single input image. It is assumed that most images used for representation learning will not have labels or other task-relevant metadata. Thus, the context and target are typically both set to the original image, after which they may be augmented in different ways. For example, in a Variational Autoencoder (VAE) [9], an input image (context) is encoded into a vector representation and then decoded back into pixels, which is then compared against the same input image (now interpreted as a target).

Once we move to sequential decision-making, the observations have a sequential structure, and there is a notion of actions and a reward function. These can all be leveraged in the construction of the contexts and targets. For example, a *Temporal VAE* is identical to a regular VAE, except that for a context observation $o_t$, we set the target to be a future observation $o_{t+k}$. Now, the input observation $o_t$ (context) is encoded into a vector representation and then decoded back into pixels, which is then compared against the *future* observation $o_{t+k}$ (target). By using a temporal target, we now incentivize representations that contain *predictive* information [20]. In reinforcement learning, another option is to add the reward $r_t$ to the target to encourage learning representations that are useful for planning.

## 2.2 Loss type

We divide modern methods for representation learning into four categories:

**Reconstruction.** Here, the goal is to reconstruct the target $y$ from the representation $z$. Both the VAE and temporal VAE in the previous section use a reconstructive loss, in which a *decoded image* $d_\phi(z)$ is compared against the target $y$, and that reconstruction loss is combined with a regularization term.

**Contrast.** Contrastive methods take a series of context–target pairs $(x_1, y_1), (x_2, y_2), \ldots, (x_K, y_K)$ and use the same network to encode both the context and target into latent representations $z_i \sim e(z \mid x_i)$ and $z_i' \sim e(z \mid y_i)$. A contrastive loss then incentivizes $z_i$ and $z_i'$ to be similar to each other, but different from $z_j$ and $z_j'$ for all other pairs $j \neq i$. Typically, the contrastive loss function is chosen to maximize the mutual information $I(z; y)$, such as with the InfoNCE loss function [21]:

$$\mathcal{L}_{\text{InfoNCE}} = \mathbb{E}\left[\log \frac{e^{f(x_i, y_i)}}{\frac{1}{K}\sum_{j=1}^{K} e^{f(x_i, y_j)}}\right]$$

$f$ could, for instance, be a bilinear function $f(x_i, y_i) = z_i^T W z_i'$, where $z_i \sim e(z \mid x_i)$, $z_i' \sim e(z \mid y_i)$, and $W \in \mathbb{R}^{n \times n}$ is a learned parameter matrix.

**Bootstrapping.** This is a simplified variant of contrastive learning. Given a related context $x$ and target $y$, a bootstrapping method predicts a moving-average-encoded target from the encoded context. Bootstrapping does not need a large dataset of negatives to prevent the representation from collapsing to a single point; instead, it prevents collapse by stopping gradients from propagating through the target encoder.

**Consistency.** These methods, such as AugMix [10] and FixMatch [11], include auxiliary loss terms that encourage the model to produce similar distributions over $y$ for different transformations of the same input image.

**Compression.** A representation $z \sim e_\theta(\cdot \mid x)$ should contain enough information about the input $x$ to solve downstream tasks. Ideally, $e_\theta$ should also extract only the *minimum* amount of information about $x$ that is necessary to perform well. We refer to this as *compression*. As a form of explicit compression, we implement the *conditional entropy bottleneck* (CEB) [22], which approximately minimizes $I(X; Z \mid Y)$.

## 2.3  Augmentation

In many algorithm designs, one or both of the context frame and target frame undergo augmentation before being processed by the encoder and decoder networks. In some algorithms, like SimCLR, this augmentation is the main source of noise causing transformed representations of the same input to not be purely identical. In other algorithms, it simply helps promote generalization by sampling from a wider image distribution than would be done naturally.

## 2.4  Neural network

In the case of a VAE, the neural network consists of two parts. The *encoder* produces the latent representation from the input, while the *decoder* reconstructs the input from the latent representation. We generalize this terminology and *define* the encoder for an arbitrary RepL method to be that part of the neural network that is used to compute the representation, and the decoder to be the rest of the neural network. Under this definition, the downstream tasks (which could include imitation, classification, reinforcement learning, etc.) only require the encoder, not the decoder. Note that the "decoder" may not convert the learned representation into some human-interpretable format; it is simply those parts of the neural network that are required by the RepL method but that do not serve to compute the representation.

### 2.4.1  Encoder

The encoder is the core component of a representation learner: it is responsible for mapping input targets $x$ into $z$ vectors that are used as the learnt representation in downstream tasks.

**Recurrent encoders.** In some cases, a "context" could be a sequence of frames instead of a single frame, and the encoder could compress that into a single representation of the past. This paper doesn't address recurrent encoders, opting instead to make all encoders operate on single framestacks.

**Momentum encoders.** In contrastive tasks, learning a high-quality representation often requires large batch sizes, since the difficulty of the contrastive task scales with the number of negatives.

However, batches of the appropriate difficulty can be so large that encoding the negative targets becomes prohibitively compute- and memory-intensive. He et al. [13] propose reusing negative targets from previous batches to alleviate this cost. One challenge with reusing targets is that the encoder can change too quickly during training, in which case negative targets from previous batches become "stale". Thus He et al. [13] use a separate *target encoder* which is updated slowly enough that targets do not become stale too quickly. Specifically, the target encoder's weights $\theta_t$ are updated to track the main context encoder weights $\theta_c$ using the update rule $\theta_t \leftarrow \alpha \theta_t + (1 - \alpha)\theta_c$. $\alpha$ is referred to as a *momentum* parameter, and is typically set to some value close to 1 (e.g. $\alpha = 0.999$).

### 2.4.2 Decoder

Decoders are optional neural network layers applied before a loss is calculated, but which are *not* included in the learnt encoder used at transfer time. They take in the $z$ output by the encoder (and optional additional information), and produce an input to the loss function.

**Image reconstruction.** The most common historical form of decoder in a RepL algorithm is the image reconstruction decoder, which has historically been used by VAEs and similar model designs to "decode" a predicted image from a representation bottleneck. This predicted image is used in calculating a MLE loss against the true image, but is discarded before downstream transfer tasks.

**Projection heads.** Projection heads are multi-layer perceptrons that take in the output of the encoder and project it into a new space over which the loss can then be calculated. Recent work has shown these to be useful for contrastive learning [2].

**Action conditioning.** Temporal tasks can be made easier by conditioning on the action $a_t$. However, for an encoder to be used for reinforcement learning or imitation, the representation must not depend on the current action $a_t$. Thus, the encoder is only responsible for learning a $z$ representation of the observation $o_t$, and is combined with a representation of the action within the decoder step.

### 2.5 Pretraining vs joint training

Another question is how to integrate representation learning with an RL algorithm. In image recognition, representation learning is done as a pretraining step. We experiment with this approach in this work, as well as the strategy of "joint training", where we add the representation learning loss as an *auxiliary loss* while performing reinforcement learning.

## 3 Experiments

Given our framework, it is straightforward to construct RepL algorithms that differ along any of the axes described in Section 2. In this section, we create a representative set of such algorithms and evaluate various ways of combining them with imitation learning. Although some RepL methods appear to be effective on some tasks, we find that the difference between using and not using RepL is often much less than the difference between using and not using augmentations for the imitation policy. In Section 4, we discuss possible reasons why RepL does not have a greater effect, and suggest alternative ways that RepL could be used more fruitfully.

### 3.1 Experiment setup

**Environments and training data.** We evaluate on ten tasks taken from three benchmark domains: DMC [8], Procgen [7], and MAGICAL [6]. Here we briefly explain our choice of tasks and datasets; for more detailed information (e.g. dataset sizes and collection methods), refer to Appendix D.

From DMC, we take image-based versions of the cheetah-run, finger-spin, and reacher-easy tasks. All three of these are popular benchmark tasks for deep RL and deep IL, and represent a range of difficulties (reacher-easy being the easiest, and cheetah-run being the hardest). However, they provide limited evaluation of generalisation. We use a common demonstration set for RepL and IL.

From Procgen, we choose the "easy" variants of the CoinRun, Fruitbot, Jumper and Miner tasks. In Procgen, different random initialisations for a given task can have wildly different appearance and structure, but still admit a common optimal policy. This makes it a much more challenging evaluation of generalization than DMC. As with DMC, we use the same demonstration set for RepL and IL.

Table 3: Design decisions for representation learning algorithms used in our experiments.

| Algorithm | Task | Context | Target | Act | Aug |
|---|---|---|---|---|---|
| Temporal CPC | Contrastive | $o_t$ | $o_{t+1}$ | ✗ | ✓ |
| SimCLR | Contrastive | $o_t$ | $o_t$ | ✗ | ✓ |
| VAE | Reconstructive | $o_t$ | $o_t$ | ✗ | ✗ |
| Dynamics | Reconstructive | $o_t, a_t$ | $o_{t+1}$ | ✓ | ✗ |
| Inverse Dynamics | Reconstructive | $o_t, o_{t+1}$ | $a_t$ | ✗ | ✗ |

From MAGICAL we choose the MoveToRegion, MoveToCorner, and MatchRegions tasks, which represent a range of difficulty levels (MoveToRegion being the easiest, and MatchRegions being the hardest). For each task, MAGICAL defines a "demo variant" for training and a set of "test variants" for evaluating robustness to changes in dynamics, appearance, etc. Unlike DMC and Procgen, our MAGICAL experiments augment the demonstration set with additional demo variant random rollouts for RepL training. This models the setting in which it is cheap to collect additional data for self-supervised learning, but expensive to collect demonstrations. We include more detailed environment setups in Appendix B.

**Imitation baselines.** Most of our experiments use behavioral cloning (BC) [23] as the base imitation learning algorithm. Given a dataset $\mathcal{D} = \{(x_0, a_0), (x_1, a_1), \ldots\}$ of observation–action tuples drawn from a demonstrator, BC optimises the policy $\pi_\theta(a \mid x)$ to maximise the expected log likelihood,

$$\mathcal{L}(\theta) = \mathop{\mathbb{E}}_{(x,a)\sim\mathcal{D}} \left[ \log \pi_\theta(a \mid x) \right].$$

We combine BC with representation learning in two ways. First, we use RepL to pretrain all but the final layer of the policy, then fine-tune the policy end-to-end with BC. This appears to be the most popular approach in the vision literature. Second, we use RepL as an auxiliary objective during BC training, so that both imitation and representation learning are performed simultaneously. Importantly, we also do control runs both with and without image augmentations. The deep RL community has repeatedly found that image augmentations can yield a greater improvement than some sophisticated representation learning methods [18, 5], and so it is important to distinguish between performance gains due to the choice of RepL objective and performance gains due to the use of augmentations.

In addition to BC, we present results with Generative Adversarial Imitation Learning (GAIL) [24] and RepL pretraining. GAIL treats IL as a game between an imitation policy $\pi_\theta(a \mid x)$ and a discriminator $D_\psi(x, a)$ that must distinguish $\pi_\theta$'s behaviour from that of the demonstrator. Using alternating gradient descent, GAIL attempts to find a $\theta$ and $\psi$ that attain the saddle point of

$$\max_\theta \min_\psi \left\{ - \mathop{\mathbb{E}}_{(x,a)\sim\pi_\theta} \left[ \log D_\psi(x, a) \right] - \mathop{\mathbb{E}}_{(x,a)\sim\mathcal{D}} \left[ \log(1 - D_\psi(x, a)) \right] + w_H H(\pi_\theta) \right\}.$$

Here $H$ is an entropy penalty weighted by regularisation parameter $w_H \geq 0$. We use augmentations only for the GAIL discriminator, and not the policy (we could not get GAIL to train reliably with policy augmentations). Discriminator regularisation is of particular interest because past work has shown that discriminator augmentations are essential to obtaining reasonable imitation performance when applying GAIL to image-based environments [25]. For our experiments combining GAIL with RepL, we use the learned representation to initialize both the GAIL discriminator and the GAIL policy.

**RepL algorithms.** Using our modular representation learning framework, we construct five representation learning algorithms described in Table 3. More detailed descriptions are in Appendix B.

### 3.2 Results

Results are shown in Table 4 for BC + RepL pretraining, and Table 5 for BC + RepL joint training, and Table 6 for GAIL + RepL pretraining. Each cell shows mean $\pm$ standard deviation over at least five random seeds. We treat IL with augmentations (but no RepL) as our baseline. We color cells that have a higher mean return than the baseline, and mark them with an asterisk (*) when the difference is significant at $p < 0.05$, as measured by a one-sided Welch's t-test without adjustment for multiple comparisons. We include the loss curves for our BC experiments in Appendix H.

Table 4: Pretraining results for BC. We color cells that have a higher mean return than BC with augmentations, and mark them with an asterisk (*) when the difference is significant at $p < 0.05$, as measured by a one-sided Welch's t-test without adjustment for multiple comparisons.

| Env | Task | Dynamics | InvDyn | SimCLR | TemporalCPC | VAE | BC aug | BC no aug |
|---|---|---|---|---|---|---|---|---|
| DMC | cheetah-run | 482±36 | 669±18 | 687±17 | 661±13 | 458±39 | 690±17 | 617±34 |
|  | finger-spin | 718±17 | 748±17* | 726±1 | 723±4 | 751±6* | 730±9 | 940±4* |
|  | reacher-easy | 774±24 | 890±14 | 907±9 | 893±13 | 880±20 | 874±21 | 452±34 |
| Procgen | coinrun-train | 8.1±0.4 | 8.0±0.2 | 8.0±0.5 | 8.1±0.3 | 8.4±0.4 | 8.1±0.3 | 8.7±0.6* |
|  | fruitbot-train | 3.2±1 | 16.2±1.2 | 17.5±1.9 | 15.4±1.5 | 17.5±1.5 | 18.3±1.9 | 11.4±0.6 |
|  | jumper-train | 8.1±0.2 | 8.0±0.4 | 7.9±0.6 | 7.5±0.6 | 7.9±0.6 | 8.1±1.2 | 7.1±1.2 |
|  | miner-train | 4.5±1.2 | 5.9±0.2 | 9.9±0.4 | 9.5±2.3 | 10.4±0.3* | 9.8±0.3 | 8.1±0.3 |
|  | coinrun-test | 6.3±0.8 | 6.9±0.5 | 6.8±0.5 | 6.8±0.4 | 7.0±0.5 | 6.7±0.4 | 6.5±0.7 |
|  | fruitbot-test | -3±0.9 | 15.6±1.1 | 13.4±1.0 | 14.7±1.0 | 13.2±1.0 | 13.7±1.1 | 2.2±0.6 |
|  | jumper-test | 3.2±0.4 | 3.9±0.3 | 3.6±0.4 | 3.7±0.5 | 3.4±0.5 | 3.9±0.5 | 4.6±0.4 |
|  | miner-test | 0.6±0.1 | 2.6±0.1 | 2.6±0.4 | 3.1±0.4 | 2.7±0.3 | 2.7±0.4 | 0.8±0.1 |
| MAGI-CAL | MatchRegions | 0.42±0.04 | 0.42±0.04 | 0.42±0.03 | 0.41±0.01 | 0.42±0.03 | 0.43±0.02 | 0.28±0.08 |
|  | MoveToCorner | 0.84±0.07 | 0.83±0.04 | 0.83±0.04* | 0.80±0.02 | 0.78±0.06 | 0.78±0.05 | 0.72±0.04 |
|  | MoveToRegion | 0.82±0.02* | 0.83±0.02* | 0.82±0.01* | 0.81±0.01* | 0.81±0.05* | 0.74±0.02 | 0.81±0.04* |

Table 5: Joint training results for BC. We color cells that have a higher mean return than BC with augmentations, and mark them with an asterisk (*) when the difference is significant at $p < 0.05$, as measured by a one-sided Welch's t-test without adjustment for multiple comparisons.

| Env | Task | Dynamics | InvDyn | SimCLR | TemporalCPC | VAE | BC aug | BC no aug |
|---|---|---|---|---|---|---|---|---|
| DMC | cheetah-run | 723±14* | 716±23* | 717±11* | 716±16* | 724±12* | 690±17 | 617±34 |
|  | finger-spin | 755±6* | 755±12* | 732±15 | 725±12 | 755±3* | 730±9 | 940±4* |
|  | reacher-easy | 898±19 | 903±10* | 889±14 | 912±18* | 903±8* | 874±21 | 452±34 |
| Proc-gen | coinrun-train | 8.0±0.4 | 7.1±0.3 | 8.0±0.5 | 8.6±0.5* | 7.9±0.2 | 8.1±0.3 | 8.7±0.6* |
|  | fruitbot-train | 17.0±0.7 | 6.6±1.4 | 13.4±1.9 | 11.4±0.7 | 15.4±1.0 | 18.3±1.9 | 11.4±0.6 |
|  | jumper-train | 7.9±0.5 | 8.1±0.4 | 8.0±0.4 | 8.0±0.3 | 8.3±0.5 | 8.1±1.2 | 7.1±1.2 |
|  | miner-train | 8.9±0.8 | 8.9±0.7 | 8.7±0.3 | 7.1±0.8 | 8.6±0.7 | 9.8±0.3 | 8.1±0.3 |
|  | coinrun-test | 6.4±0.4 | 6.0±0.5 | 6.6±0.3 | 6.2±0.5 | 6.9±0.4 | 6.7±0.4 | 6.5±0.7 |
|  | fruitbot-test | 10.9±0.7 | 3.3±1.1 | 8.5±1.5 | 6.4±1.2 | 10.4±1.6 | 13.7±1.1 | 2.2±0.6 |
|  | jumper-test | 3.4±0.3 | 4.8±0.2* | 3.8±0.3 | 3.4±0.3 | 3.9±0.7 | 3.9±0.5 | 4.6±0.4* |
|  | miner-test | 2.0±0.2 | 1.9±0.3 | 1.8±0.3 | 1.0±0.2 | 2.0±0.3 | 2.7±0.4 | 0.8±0.1 |
| MAGI-CAL | MatchRegions | 0.44±0.02 | 0.23±0.08 | 0.41±0.02 | 0.01±0.01 | 0.41±0.03 | 0.43±0.03 | 0.31±0.02 |
|  | MoveToCorner | 0.78±0.07 | 0.30±0.22 | 0.76±0.05 | 0.02±0.02 | 0.82±0.06 | 0.80±0.05 | 0.70±0.09 |
|  | MoveToRegion | 0.76±0.02 | 0.35±0.24 | 0.74±0.01 | 0.47±0.07 | 0.77±0.02 | 0.75±0.02 | 0.78±0.04 |

**BC pretraining results.** In the pretraining setting, we see that none of our RepL algorithms consistently yield large improvements across all (or even most) tasks. Indeed, the relative impact of adding representation learning tends to be lower than the impact of adding or removing augmentations. Although adding augmentations to BC usually yields a large improvement, there are a handful of tasks where adding augmentations substantially decreases performance; we remark further on this below. Note that most of our RepL algorithms do seem to yield an improvement in MoveToRegion, suggesting that there may still be value to RepL for a narrower set of tasks and datasets.

**BC joint training results.** When using joint training as an auxiliary loss, we similarly see that no one RepL method consistently improves performance across all benchmark tasks. However, in the DMC tasks, we do see consistent improvement over the baseline for all RepL methods. This suggests that our RepL methods provide benefit in some environments, but are sensitive to the choice of task.

**Effect of augmentations on BC.** Incorporating augmentations into BC training tended to yield the largest effect of any technique considered in this work, even without an explicit representation learning loss. In roughly half of the environments studied, this had a substantial impact on reward, and reward increased 150% or more in reacher-easy, Fruitbot, and MatchRegions. However, environments seem to be bimodal in their response to augmentations: in a handful of environments (finger-spin, coinrun-train, jumper-test, and MoveToRegion), adding augmentations leads to consistently *worse*

Table 6: Pretraining results for GAIL. We color cells that have a higher mean return than BC with augmentations, and mark them with an asterisk (*) when the difference is significant at $p < 0.05$, as measured by a one-sided Welch's t-test without adjustment for multiple comparisons. For the sake of space, we abbreviate TemporalCPC to $t$CPC.

| Env | Task | Dynamics | InvDyn | SimCLR | $t$CPC | VAE | GAIL aug | GAIL no aug |
|---|---|---|---|---|---|---|---|---|
| DMC | cheetah-run | 380±76 | 320±61 | 265±58 | 360±74 | 375±33 | 449±67 | 75±40 |
| | finger-spin | 868±14 | 886±8* | 800±23 | 748±72 | 868±18 | 868±12 | 0±0 |
| | reacher-easy | 53±24 | 73±51 | 21±23 | 118±88 | 122±89 | 221±162 | 89±88 |
| Proc-gen | coinrun-train | 5.9±0.29* | 5.85±0.51* | 2.15±1.53 | 3.28±2.62 | 3.54±1.22 | 3.31±0.44 | 2.80±0.89 |
| | fruitbot-train | -2.81±0.1 | -2.37±0.55 | -2.47±0.15 | -2.38±0.31 | -2.49±0.22 | -2.42±0.42 | -2.63±0.30 |
| | jumper-train | 3.31±0.31 | 3.17±0.40 | 3.36±0.53 | 2.69±1.31 | 3.45±0.70 | 3.44±0.52 | 3.47±0.53 |
| | miner-train | 0.53±0.12 | 0.60±0.11 | 0.53±0.14 | 0.84±0.14* | 0.51±0.07 | 0.65±0.10 | 0.77±0.18 |
| | coinrun-test | 6.1±0.9* | 5.91±0.16* | 2.11±1.61 | 3.35±2.74 | 3.01±1.10 | 3.44±0.68 | 2.77±0.84 |
| | fruitbot-test | -2.44±0.49 | -2.65±0.24 | -2.55±0.30 | -2.65±0.14 | -2.85±0.33 | -2.44±0.50 | -2.51±0.44 |
| | jumper-test | 2.56±0.52 | 2.53±0.64 | 3.15±0.45 | 2.35±0.81 | 2.75±0.59 | 3.25±0.42 | 3.15±0.20 |
| | miner-test | 0.36±0.04 | 0.57±0.07 | 0.55±0.24 | 0.87±0.15* | 0.50±0.17 | 0.65±0.17 | 0.66±0.14 |
| MAGI-CAL | MatchRegions | 0.42±0.10 | 0.34±0.12 | 0.47±0.04 | 0.39±0.12 | 0.30±0.15 | 0.46±0.06 | 0.22±0.12 |
| | MoveToCorner | 0.48±0.09 | 0.45±0.10 | 0.52±0.07 | 0.55±0.15 | 0.62±0.11* | 0.49±0.08 | 0.55±0.14 |
| | MoveToRegion | 0.72±0.07 | 0.74±0.04 | 0.74±0.06 | 0.76±0.03 | 0.75±0.07 | 0.75±0.09 | 0.60±0.14 |

performance. This effect is particularly dramatic in finger-spin, which we believe is a result of the fact that relevant objects in the environment always stay fixed. Consequently, translational augmentations don't aid generalization, and rotational augmentations can be confused with true signal (since the angle of the finger determines the ideal action). Because augmentation already yields large benefits, many of the representation learning algorithms do not provide much additional gain on top of BC-Augs, even when they perform substantially better than BC-NoAugs. This result is consistent with the finding by Laskin et al. [4] that simply augmenting input frames in reinforcement learning produced performance on par with sophisticated representation learning methods.

**GAIL pretraining results.** GAIL pretraining results mirror those for BC pretraining, but with even fewer statistically significant deviations from baseline performance. We see that augmentation can be even more important for GAIL than it is for BC. For instance, GAIL with discriminator augmentations obtains higher return on finger-spin than BC does, but obtains a return of 0 when discriminator augmentations are removed. This is consistent with the observation of Zolna et al. that strict regularisation is essential to make GAIL perform well in image-based domains [25].

## 4 Discussion & future work

**Contrasting image classification and imitation learning datasets.** The use of self-supervised representation learning for pretraining has met with notable success in image classification [2]. By comparison, results from RL literature have been mixed, with some positive results, but also several works [4, 5] which claim that RepL adds little value relative to image augmentation—a result which we observe in imitation as well. Given this, it's natural to wonder *why* successes from supervised learning have not been reproduced in sequential decision making problems such as RL and imitation.

The case of Behavioural Cloning (BC) is particularly illustrative. BC uses the same optimization algorithms, loss types, and network architectures as other forms of image classification, so if RepL is less helpful for BC than for other forms of classification then it must be due to the choice of training and evaluation data. For the sake of illus-

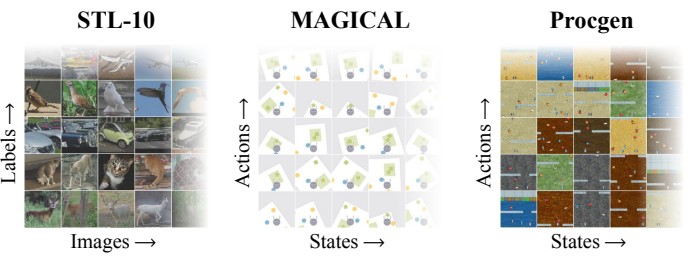

Figure 2: We show a sample of STL-10, MAGICAL, and Procgen images. Images on the same row have the same label (bird, car, etc.) or expert action (up, down, etc.). It can be easier to tell whether two images have the same label in classification than in IL tasks.

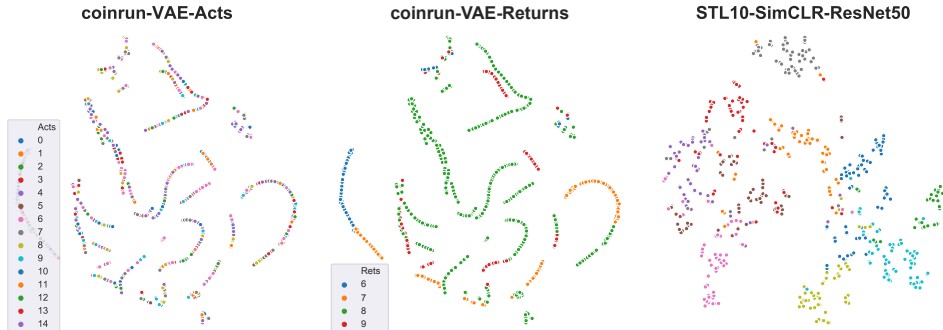

Figure 3: t-SNE embedding of representations from a VAE encoder on CoinRun, labeled with the corresponding actions (left) and discretized returns (middle). Returns are estimated by applying GAE to an expert PPO demonstrator, then discretized by rounding to the nearest whole number to produce a "label". We compare these with STL-10 image representations generated by a ResNet50 pretrained with SimCLR, colored by class (right).

trating differences in data distributions, Fig. 2 compares the STL-10 dataset—a typical image classification task—with datasets for MAGICAL and Procgen. dm_control is not pictured because it has a continuous action space, so there was not a natural separation of images by action along the $y$ axis.

One notable difference in Fig. 2 is that there is less between-class variation in MAGICAL and Procgen than in STL-10: the choice of action is often influenced by fine-grained, local cues in the environment, rather than the most visually salient axes of variation (background, mean color, etc.). For example, in MAGICAL the sets of states that correspond to the "forward" and "left" demonstrator actions cover a similar visual range. Indeed, the agent's choice between "left" and "right" could change if its heading shifted by just a few degrees, even though this visual change would not be obvious to a human. In contrast, STL-10 exhibits substantial between-class variation: it's hard to confuse a the sky-blue background and metal texture of a plane for the natural setting and fur of a deer. Thus, a RepL method that simply captures the most visually salient differences between classes may be much more useful for classification on STL-10 than for control on MAGICAL or Procgen.

**What is the right downstream prediction target?** Both GAIL and BC attempt to learn a policy that predicts expert actions from observations. We've argued that RepL algorithms may be focusing primarily on the most visually salient differences between states, at the expense of the fine-grained features necessary for action prediction. However, it could be that reward- and value-prediction benefit more from a representation that captures mostly coarse-grained visual differences. Moreover, Yang and Nachum [26] have observed that state-based (as opposed to image-based) offline Q-learning *does* benefit from existing RepL techniques, even though state-based BC does not. Together, these facts suggest existing RepL methods might be more helpful when the downstream prediction target is value or reward rather than action.

To explore this hypothesis, we visualize how well RepL-learned representations align with action labels, estimated expert returns, and trajectory IDs. In Figure 3 we show t-SNE projections of observation embeddings taken from seven expert CoinRun demonstrations. The embeddings were generated by a VAE-pretrained encoder. We compare these with t-SNE clusters generated from a ResNet50 with SimCLR on ImageNet, then evaluated on STL-10 (a resized ImageNet subset).

Representations from a well-trained encoder should cluster nicely according to the label (e.g. classes, actions) used for the downstream task. We see this with the STL-10 embeddings, which cluster nicely by class. In contrast, we see that our encoders for CoinRun do not produce embeddings that cluster nicely by action. However, they do seem to cluster readily by estimated expert returns. This is likely a consequence of the events that cause states to have high value—such as being close to the far wall with the coin—depend primarily on coarse-grained features of the state. We speculate that this is likely true in MAGICAL, too, where the reward function tends to depend only on salient features like whether the agent is overlapping with any of the colored goal regions.

Our negative results for GAIL and RepL provide reason to be cautious about our conjecture that reward functions (and value functions) are more amenable to RepL. A GAIL discriminator is similar to a reward function, but the overall performance of GAIL does not change much when pretraining

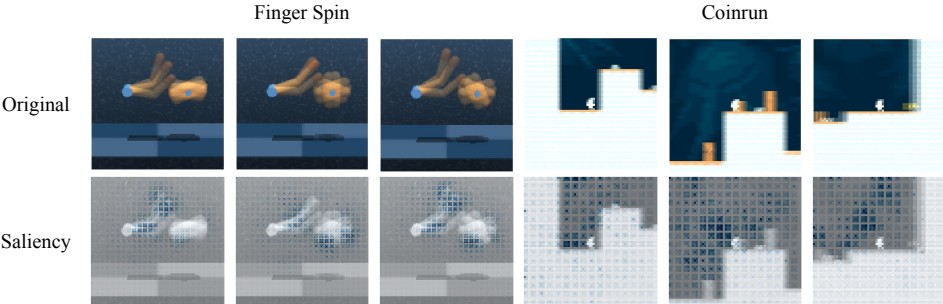

Figure 4: Saliency map generated by an encoder trained using SimCLR. Top row shows input frames, averaged across a three-frame stack of inputs. Bottom row shows saliency map overlaid on top of grayscale images, with darker blue shading indicating greater influence over the network's output. Notice that SimCLR attends mainly to the foreground in DMC, and mainly to the background in CoinRun.

the discriminator with RepL. On the other hand, it is worth noting that the GAIL discriminator does not in general converge to a valid reward function for the task, so this is not a direct test of the hypothesis that reward learning is more amenable to RepL pretraining than policy learning. We therefore believe it is still worth investigating whether imitation learning algorithms that directly learn reward functions [27] or value functions [28] benefit more from RepL than algorithms that learn policies.

**The importance of using diverse benchmark tasks.** Our experiment results in Table 5 showed much greater benefit for RepL on DMC than on Procgen and MAGICAL. This underscores the importance of evaluating across multiple benchmarks: had we only used DMC, we might have erroneously concluded that RepL is typically helpful for BC.

The finger-spin (DMC) and CoinRun (Procgen) tasks provide a useful illustration of how differences in performance across tasks can arise. Fig. 4 shows example saliency maps [29] generated by SimCLR-pretrained encoders in these two tasks. In finger-spin, the SimCLR encoder mostly attends to foreground objects, while in CoinRun it attends to the background. This makes sense: the boundary between the background and terrain is easy to detect and shifts rapidly as the agent moves, so paying attention to the shape of background is quite helpful for distinguishing between frames. Unfortunately, semantically important foreground features in CoinRun, such as obstacles and gold, are less discriminative, which is why we believe SimCLR is not dedicating as much model capacity to them. In contrast, the background in finger-spin changes very little, so SimCLR is forced to attend to foreground objects that change position between frames.

More generally, we believe that differences between RepL performance across tasks are due to implicit assumptions that our (unsupervised) RepL algorithms make about what kinds of features are important. For tasks that do not match these assumptions, the representation learning algorithms will do poorly, regardless of how much data is available. In our SimCLR example, information about background shapes crowds out task-relevant cues like the distance between the agent and an obstacle. It is therefore important for future research to (1) consider whether the implicit assumptions underlying a given RepL algorithm are likely to help models acquire useful invariances for the desired tasks; and (2) test on multiple domains to ensure that the claimed improvements are robust across environments.

## 5 Conclusion

We have seen that, when compared against a well-tuned IL baseline using image augmentations, the impacts of representation learning for imitation are limited. On some benchmark suites it appears that it helps, while on others there is not much impact, suggesting that the effect of RepL is quite benchmark-specific. Our analysis has identified several hypotheses that could help understand *when* and *where* representation learning can be useful. We are excited to see future work investigate these hypotheses, and hope the EIRLI framework can serve as a useful starting point for any such investigations.

## Acknowledgments and Disclosure of Funding

The authors would like to thank Michael Chang, Ben Eysenbach, Aravind Srinivas, and Olivia Watkins for feedback on earlier versions of this work. This work was supported in part by the DOE CSGF under grant number DE-SC0020347, along with a grant from the Open Philanthropy Project and computational support from Google.

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
