# OpenReview forum: "An Empirical Investigation of Representation Learning for Imitation"
_NeurIPS.cc/2021/Track/Datasets_and_Benchmarks/Round2 — NeurIPS 2021 Datasets and Benchmarks Track (Round 2)_

### Official Review · Reviewer_nHUv · 2021-09-20
**A modular framework for the construction and evaluation of representation learning algorithms**

**Rating:** 6
**Confidence:** 3

**Strengths:**

A Modular framework from which varying implementations of representation learning can be constructed, allowing for effects of various design decisions to be analyzed.
An analysis of various representation learning algorithms compared to behavioral cloning with and without augmentation, that reinforce previous work that shows little additional improvement in using representation learning over augmentation.

**Weaknesses:**

All the benchmarks that were used relied on synthetic data. In Sections 4.2 and 4.3 the variation in the datasets and the importance of evaluating across different benchmarks is discussed and STL10 is mentioned. Including a benchmark that uses real world data would add value.


**Additional Feedback:**

I'm not sure if including the implementation sections were needed.
Figures 2 and 4 were a little hard to read and in Figure 4 showing the original scenes would help give more context.
In the conclusion the variation in datasets is discussed and, having figure 3 show samples from the DMC task would be helpful.
It would be useful to see the results of the preliminary experiments in the Appendix that showed momentum, projection heads, and compression did not show any advantage.


**Clarity:**

The paper was clearly written and easy to follow.


**Correctness:**

The evaluation methods and experiments were appropriate and performed correctly.

**Documentation:**

The Github repository linked has the code and instructions to reproduce the results and has a link to download the dataset that was used.


**Ethics:**

There were not any ethical concerns.


**Relation To Prior Work:**

The paper provides an analysis of different representation learning approaches for imitation learning in image-based environments and the design choices that were made (target selection, loss type, use of augmentation, neural network encoder/decoder type, and whether it is used for pretraining or joint training).


**Summary And Contributions:**

The authors argued that most common representation learning algorithms for imitation learning vary over the choice of target selection, loss type, use of augmentation, neural network encoder/decoder type, and whether it is used for pretraining or joint training. From this framework  a code base was constructed that can be used for building Representation Learning algorithms. This was used to evaluate the utility of representation learning over 10 tasks from 3 benchmarks (MAGICAL, Procgen, DeepMind Control Suite) compared with Behavioral Cloning with and without data augmentation. Representation learning was able to outperform data BC with augmentation on the DMC benchmarks but had little impact on MAGICAL or Procgen.

---

> ### Author Response · Authors · 2021-09-30
> **Response to reviewer nHUv**
>
> Thank you very much for your helpful feedback and recognition of our work. We appreciate that you think our paper is valuable in (1) its modular framework, (2) our results and analysis on various representation learning algorithms. We address your comments and questions in the below.
>
> **Q1. Results with real world datasets.**
>
> **A1.** Thank you for your suggestions. We are unsure whether you are referring to environments with labels generated by actual humans, or real-world image environments in your comments on “real-world data”, so we will respond to both.
>
> If you mean behavioral realism (i.e. demonstrations given by humans) - while ProcGen and DMC involve synthetic datasets, MAGICAL does not. In MAGICAL, the demonstrations were provided by humans, rather than “experts” trained with RL. We do not see obvious differences in our results that set MAGICAL apart from Procgen, so we do not currently expect significant differences from synthetic vs. non-synthetic datasets, though of course not much can be concluded from results on one human and two synthetic benchmarks.
>
> If you mean visual realism - As noted in our response to Review zMjo, we have attempted to add CARLA results to the paper by using the existing imitation dataset distributed with D4RL.  However, we have not been able to tune and train a near-expert policy during the rebuttal. In our preliminary experiments, BC with augmentation obtained a mean return of \~13 (slightly better than random), and SimCLR & VAE pretrained policies are <1 (\~random). Given that these figures are much lower than those in the existing literature, we chose not to include them in the latest revision. However, we will continue tuning this method and endeavour to include full CARLA results in the camera-ready.
>
> While we agree that experiments on more visually realistic environments would be illuminating (as in our response to Reviewer zMjo), we want to emphasize that our current results are also quite interesting. Our expectation when starting this project was that we would see benefits from representation learning across all of our benchmarks. After all, most of the papers combining representation learning and reinforcement learning (see Table A1) report positive results on environments similar to the ones we use even though they are usually not visually realistic. We therefore expected that the effect on imitation learning should be similar, which turned out not to be the case.
>
> **Q2. Typos & additional feedback.**
>
> **A2.** We have enlarged Figures 2 & 4, and included original scenes in Figure 4. We left DMC out of Figure 3 because its actions are continuous, so it is not clear what the y-axis should be in that figure (in the other figures it is either the label or the discrete action). We’ve included experiments for projection heads, compression, and momentum in Appendix J.

---

### Official Review · Reviewer_zMjo · 2021-09-20
**Interesting work in representation learning for imitation but could benefit from more experimentation**

**Rating:** 6
**Confidence:** 3
**Clarity:** The paper is written clearly.

**Strengths:**

1. Interesting idea: I found the idea of testing the usefulness of RepL in imitation learning pretty interesting. Although deep RL has found a lot of traction over the past few years and still offers a lot of scope for improvement and progress,  imitation learning is well-known to work much better and scale in real-world scenarios for control tasks. In my view, trying to improve policies learned through imitation by training good representations is an important line of work and deserves more attention in current and future research.

2. Generality of proposed framework: The codebase for designing customized RepL algorithms looks pretty generic and easily extendable. It could serve as a useful tool for designing more advanced imitation learning specific RepL algos. in the future.

3. Somewhat comprehensive experimentation: The paper does a decent job at comparing different RepL-based imitation techniques across multiple tasks and environments of varying complexity. That helps the authors make observations about how simple data augmentation fares against more sophisticated RepL algorithms in such settings. It also helps the authors identify the kind of choices that one should generally make in designing their own RepL algorithm.

**Weaknesses:**

Lack of experiments on more realistic datasets: At several points in the paper, the authors note why RepL could be expected to work not as well as in the case of more realistic vision and text data. They mention that the inherent simplicity of the data in the settings considered in the paper allows the models to solve the task without learning high-level semantic concepts. Such behavior reduces the ability of RepL to improve the task performance of the learned policies. If that's the case, it's important to extend the evaluation to more photo-realistic settings like embodied control, self-driving, etc., or even better, to robotic control in real-world environments.

P.S. -- I am aware that doing a comprehensive evaluation with an actual robot in real-world settings could be very hard during a short rebuttal period, but it would be good to see some results in more realistic simulators like CARLA, Thor, Habitat, etc.


Edit: rating updated

**Additional Feedback:**

Other questions/suggestions:
1. L52-54, 'We argue ... Fig. 1' -- the 'axes of variation' is not very clear; either change the statement or change the figure.

2. L149-150, 'We have implemented ... a RecurrentEncoder' -- as far as I understand, even momentum encoders could be single image based encoders. If that's the case, then this statement isn't entirely correct?

3. L219-220, 'Using our modular ... learning algorithms': any particular reason why you used just these 5? Also related to this, maybe report the missing experiments mentioned in L228-230 ('Note that we ... doing so') in the Supp.

4. L242, 'Note that mot' -- 'Note that most'?

5. Table 4 and 5: the table captions aren't self-explanatory. As a reader, it's not very pleasing to go back to the text to understand a table.

**Correctness:**

The claims made in the paper and the supporting experiments/evaluation methods look correct to me.

**Documentation:**

The documentation supporting the paper looks well-done.

**Ethics:**

I couldn't spot any ethical issues.

**Relation To Prior Work:**

The work compares with prior work and tries to look at a direction that's not been previously looked at. However, some more evaluation would make the work even stronger. I have listed such possible evaluation under 'Weakness'.

**Summary And Contributions:**

This work aims to study the utility of using representation learning in the process of learning control behaviors using imitation learning. The contributions of this work are two-fold:
1. It identifies different elements of variation in a standard representation learning (RepL) approach used in adjacent ML fields like NLP, computer vision, etc., and designs a generic framework for coming up with customized RepL approaches by tuning the elements of variation, for imitation-based control. The authors have built a modular codebase to that extent and open-sourced it.

2. The paper compares sophisticated RepL approaches generated using the generic framework against simple well-tuned baselines that use data augmentation, and shows that across multiple tasks and environments, data augmentation often outperforms both representation learning based pre- or joint training. It further tries to analyze the learned representations and shows a lack of correlation among representations for similar actions, thus showing that using ground truth action as the learning signal in imitation doesn't really benefit from RepL.

---

> ### Author Response · Authors · 2021-09-30
> **Response to reviewer zMjo**
>
> Thank you very much for your helpful feedback and recognition of our work. We appreciate that you think our paper is valuable in (1) our modular framework, (2) its results and analysis of the more neglected but more empirically useful algorithm, imitation learning, (3) its comprehensive experimentation, and (4) the well-documented code base. We address your comments and questions in the below.
>
> **Q1. Experiments on more realistic datasets**
>
> **A1.** Thank you for pointing this out. Per your suggestion, we have attempted to add CARLA results to the paper by using the existing imitation dataset distributed with D4RL.  However, we have not been able to tune and train a near-expert policy during the rebuttal period. In our preliminary experiments, BC with augmentation obtained a mean return of \~13 (slightly better than random), while SimCLR & VAE pretrained policies are <1 (\~random). Given that these figures are much lower than those in the existing literature, we chose not to include them in the latest revision. However, we will continue tuning this method and endeavour to include full CARLA results in the camera-ready.
>
> While we agree that experiments on more visually realistic environments would be illuminating, we want to emphasize that our current results are also quite interesting. Our expectation when starting this project was that we would see benefits from representation learning across all of our benchmarks. After all, most of the papers combining representation learning and reinforcement learning (see Table A1) report positive results on environments similar to the ones we use even though they are usually not visually realistic. We therefore expected that the effect on imitation learning should be similar, which turned out not to be the case. Further, note that imitation algorithms are far from solving the tasks in at least MAGICAL and Procgen at the demonstrator level, so the tasks are not too “simple” for repL to improve performance.
>
> **Q2. “…any particular reason why you used just these 5 [repL algorithms]?”**
>
> **A2.** We felt these five were a reasonably representative selection across a few popular axes: contrastive (SimCLR, TCPC) vs generative (inv. dynamics, dynamics, VAE); dynamics-aware (TCPC, inv. dynamics, dynamics) vs. not dynamics aware (VAE, SimCLR); state prediction (SimCLR, TCPC, dynamics, VAE) vs. action prediction (inv. dynamics), and action conditioning (dynamics) vs. no action conditioning (SimCLR, TCPC, inv. dynamics, VAE). Appendix J now contains ablations for the other design axes that we did not evaluate in the initial submission (e.g. compression, projection heads, etc.)—the results support our preliminary finding that these axes are not especially important.
>
> **Q3. Typos/clarity issues.**
>
> **A3.** Thanks for pointing out our typos and clarity issues, including the accidental implication that MomentumEncoder does not work on individual images (it does work on individual images). Our latest revision fixes these issues.

---

> > ### Comment · Reviewer_zMjo · 2021-10-01
> > **Response to rebuttal**
> >
> > Thanks for the clarifications. The authors have been able to address verbally my main concern about the lack of effectiveness of RepL on less realistic datasets. However, I would really love to see some experiments on more visually challenging datasets in future revisions of the paper. Also, I am increasing my rating for the paper.

---

### Official Review · Reviewer_bFea · 2021-09-21
**Refreshing analysis and implementation, but intuition for results unclear**

**Rating:** 6
**Confidence:** 4

**Strengths:**

1. *Significance*. It is refreshing to see an honest "negative result" of this sort. In the recent trend towards "smart" representation learning in all domains, the claim is invariably that such auxiliary objectives can improve efficiency of learning higher-level objectives, with accompanying experiments picked to align with prior knowledge. However, especially in reinforcement learning and---more recently---imitation learning, it is clear that the heterogeneity of inputs can hardly guarantee that these methods work well across the board. This paper is significant in that it is one of few "large-scale" studies that attempt to measure how things actually work in a mixed, agnostic setting.

2. *Code Quality*. The structure if very well-designed, and the code is modular and reusable---something quite rare in esp. reinforcement learning and imitation learning research. Something like this conceivably could be the basis for a lot of future benchmarking work (it would almost appear irresponsible for new methods/papers not to at least run the comparison already implemented using this framework, with it appearing to be quite easy to employ).

**Weaknesses:**

1. *Imitation Learning vs. Classification*: A major empirical point is that while representation learning can benefit image classification, the same cannot be said for image-based imitation learning. However, the only imitation baseline is behavioral cloning, which is essentially a naive supervised classification algorithm. Is this not correct? Of course, in a regular classification task the performance evaluation at test time is usually different from in imitation learning. But apart from this, why should we expect BC to be different from classification?

2. *Predicting Actions vs. Rewards*: It is stated, "One possible reason is that, to the extent that visual state mostly conveys progress through a trajectory, that state will contain more information about rewards (which are typically related to trajectory progress) than actions (which are often repeated at divergent points within a trajectory)." But a basic fact from the Bellman equation is that there is a 1-1 correspondence between the space of rewards and the space of Q-functions. So if something conveys information about rewards, it also conveys information about Q-functions, which gives everything you need to predict the action (hard-max, or soft-max of those Q-functions). So it is not clear that this hypothesis is convincing. Perhaps without further structure in the loss functions, it would be "difficult" for a black-box model to automatically infer Bellman-related relationships?

3. *Other Imitation Algorithms*: Why is BC the only method used?

4. *Confounding Variables*: It is stated, "RepL pretrained policy operating on Procgen can rely heavily on background information, but this is rarely the case in DMC". Why would this be the case, if the training set is sufficiently large (so that there should ideally be no distribution shifts in background information between train and test time)? If this is impossible to guarantee, perhaps we could use a simple form of invariant representation learning to solve this problem (i.e. to decompose the observed image into noise independent of actions, and causal factors that influence actions).

**Additional Feedback:**

The implementation and design is great. My primary concern (and I would really appreciate some thought into this) relates to my first question. The authors propose some "hypotheses" for why things turned out the way they did (i.e. no consistent improvement using representation learning over image augmentation, for BC). However, I found the analysis and diagrams to be explanations of the result itself, but not an explanation of **why** such a result is obtained. Any further insights/clarity would be good.

**Clarity:**

In general the paper is well-organized and well-written.

Typo:

- Line 265: "alignged" -> "aligned"

**Correctness:**

Broadly, the evaluation methods and experiment design seem appropriately designed.

- See above question on why BC is the only method used.

- Ideally, for this sort of "deep dive" there would be even more environments/games/tasks. But the code appears very extensible, so this is not a major gripe, just a suggestion for future development.

**Documentation:**

There appears to be sufficient detail and code documentation to support reproducibility (much more so than most existing work, at any rate).

**Ethics:**

N/A.

**Relation To Prior Work:**

To the best of my knowledge, the relationship to prior work is covered satisfactorily.

**Summary And Contributions:**

This paper presents a well-thought design (and well-documented implementation) of a modular benchmarking framework for a broad investigation into whether representation learning benefits imitation learning.

The framework and software is organized according to major axes of variation in representation learning and end-to-end algorithm design, including choice of representation learning targets, type of loss function, presence of augmentation, specifics of encoder and decoder networks, as well as pre-/joint training schemes.

A primary takeaway from the implemented study is a bit of a "negative result"---that as far as behavioral cloning is concerned (as evaluated across 10 selected tasks from 3 popular environments), representation learning does not---on average---appear to add value above and beyond what staple image-based augmentation methods can already accomplish.

---

> ### Author Response · Authors · 2021-09-30
> **Response to reviewer bFea (pt1)**
>
> Thank you very much for your helpful feedback and recognition of our work. We appreciate that you think our paper is valuable in (1) its modular framework, (2) its significance in providing an honest empirical evaluation, (3) its high-quality, well-designed, modular, and reusable code base. We address your comments and questions in the below.
>
> **Q1. Imitation learning vs. classification**
>
> **A1.** We agree that behavioral cloning is similar to classification in that their losses are essentially the same, and in Section 4.2 we discussed possible reasons why repL has succeeded in classification but not IL. Since there are no major differences in loss, algorithms, or architecture, we conjecture that a large part of this difference comes from the dataset. The main insight of Section 4.2, which we have attempted to convey in Figure 3, is that traditional datasets for image classification often have much larger between-class variation than the datasets that we are using for IL. For instance, instances of the “plane” and “bird” classes in STL10 look much more different from one another, on average, than instances of the same class. In contrast, the sets of states that correspond to “forward” and “left” demonstrator actions in MAGICAL cover a similar visual range, since the agent often alternates between taking “forward” and “left” actions over the course of a trajectory in order to reach a given point. Thus there is less of a visual distinction between the different “classes” (i.e. actions) in MAGICAL than in popular image classification benchmarks. We have reworded Section 4.2 to help readers better understand our hypothesis.

---

> ### Author Response · Authors · 2021-09-30
> **Response to reviewer bFea (pt2)**
>
>
>
> **Q2. Predicting Actions vs. Rewards**
>
> **A2.** First, we should clarify the hypothesis that we are putting forward. Our submission suggests that, for the environments we are evaluating in, states with different rewards or values are easier to visually distinguish than states with similar actions. For instance, in the MAGICAL MoveToRegion task, the ground truth reward only requires the robot to detect whether its body overlaps with the coloured square, which can be accomplished with relatively coarse-grained visual primitives (e.g. by detecting colour blobs, or looking for transitions between the colour of the square and the colour of the robot’s body). In contrast, the policy requires the robot to infer its precise orientation relative to the square and take appropriate movement actions to get closer. As noted above, the optimal action can also switch back and forward rapidly depending on the precise orientation of the robot: the decision to turn left, or move forward, or turn right can be different depending on whether the square is at a heading of +5°, 0°, or -5° from the robot, even though these states are quite similar visually. Unfortunately, a SimCLR encoder or a VAE is unlikely to pick up on these minute visual distinctions in the observation. For this reason, we speculate that repL methods might be less useful for policy learning than for, e.g., learning value or reward functions.
>
> However, the reviewer makes a good counterpoint: if it’s easy to learn rewards or values, then it should also be easy to learn policies. After all, if we recover an accurate reward function, then we should be able to recover a good imitation policy by doing planning/RL (i.e. Bellman backups) on the recovered reward function. The reviewer suggests that, to the extent that our IL and repL algorithms are not taking advantage of this, it could be that they lack the structure necessary to exploit the mapping from reward functions to optimal policies.
>
> Broadly, we agree with the reviewer’s conjecture that the examined approaches lack the structure necessary to exploit the reward → policy mapping. In particular, our approach is to use a relatively small neural network on top of the learned representations in order to predict actions. The clustering in Figure 2 demonstrates that the representations make rewards easily accessible, that is, they make it so that rewards can be predicted with the additional relatively small neural network. This is not true for actions; while in principle correct actions could be computed through planning, we should not expect that such a complex, sequential calculation can be expressed by a relatively small neural network. We could add explicit inductive biases to create repL algorithms that can plan given a reward function -- while we do not know of any such existing repL algorithms, it would be an interesting avenue for future work.
>
> We have updated Section 4.1 of the paper to make the wording and implications of our hypothesis more clear, which we hope addresses the point made by the reviewer. We’ve also updated this section to reflect our new results for GAIL. Although these results show little benefit to RepL pretraining with GAIL, it is worth cautioning that the GAIL discriminator is not directly analogous to a reward function (like a GAN, the GAIL discriminator should in theory be uniformly confused in all states once the generator reaches optimality, which means it is no longer informative as a reward function). Overall, we view the negative results on GAIL as weak evidence against this hypothesis, but significantly more work would be needed to convincingly confirm or refute it. Again, we have been careful to convey these nuances in the updated revision.
>
> **Q3. Other imitation algorithms**
>
> We have added experiments with GAIL and RepL pretraining to complement the existing BC experiments. These are in Table 6 of the main paper. We see that the results are similar to those for BC and RepL pretraining. Specifically, we see that augmentations have a substantial effect on performance; in fact, in finger-spin, removing augmentations causes the imitation policy’s return to drop from near-expert levels to random levels (which we believe is because of the regularizing effect of using augmentations with the GAIL discriminator). However, there are almost no statistically significant differences between baseline performance and the performance of algorithm variants that use RepL. Again, this suggests that new approaches will be needed to get value out of representation learning in an imitation learning context.

---

> ### Author Response · Authors · 2021-09-30
> **Response to reviewer bFea (pt3)**
>
> **Q4. Confounding variables**
>
> **A4.** Thank you for your feedback. We reworded Section 4.3 and hope this will make it clearer. The root issue is that the algorithms which we compare are all unsupervised (i.e. label-free) learning algorithms, and so they must make some implicit assumptions about the (unknown) structure of the mapping from inputs to labels. For tasks that do not match these assumptions, the representation learning algorithms will do poorly, regardless of how much data is available.
>
> As an example, SimCLR assumes that the representation (and thus the label) should be roughly the same for two augmented copies of a given image, but that it should be different for different images. In Figure 4, we see that a SimCLR pretrained encoder is more likely to focus on background in CoinRun than finger-spin. This makes sense: the boundary between the background and terrain is easy to detect and shifts rapidly as the agent moves, so paying attention to the background is quite helpful for distinguishing between frames. Unfortunately, semantically important foreground features like obstacles and gold are less discriminative, which is why we believe SimCLR is not dedicating as much model capacity to them. In contrast, the background in finger-spin changes very little, so SimCLR must instead pick up on the foreground objects that actually change position between frames. As noted in Section 4.3, this suggests that it is very important to evaluate on a wide range of benchmark tasks, since a representation learning assumption that works well on one task may be harmful in another.
>
> We agree that the literature on invariant supervised learning could be helpful to solve this problem, and incorporate it as a suggestion for future research.
>
> **Q5. Typos and other feedback.**
>
> **A5.** Thanks for pointing out our typos; we have now fixed them.
>
> **Summary.** Thank you very much for your suggestions. In summary, our rebuttal and changes have attempted to clarify our hypotheses for why repL did not significantly improve performance over BC with augmentations, even though repL appears to help in image classification. The first hypothesis is that popular image classification datasets have more between-class variation than our decision-making environments, and are thus a better fit for existing visual representation learning algorithms. The second hypothesis is that action selection depends on fine-grained visual features which are particularly difficult to learn without supervision, and that unsupervised repL may therefore be better suited to other downstream prediction targets (e.g. value, reward). Finally, we posit that assumptions which are useful for some tasks can be actively harmful for others, which makes it hard to obtain consistent benefits across many existing decision-making benchmarks.

---

### Official Review · Reviewer_265a · 2021-09-21
**solid paper**

**Rating:** 7
**Confidence:** 4
**Correctness:** The claims see correct.
**Clarity:** The paper is well written.

**Strengths:**

The setup of the benchmark is nice.  The wide range of design decisions studied is a good template for future work.  The setup enables finding conclusions such as the value of data augmentation.

**Weaknesses:**

I didn't seen any major weaknesses.

Perhaps the biggest question is how the results generalize to other imitation learning approaches beyond behavioral cloning, but I don't count that as a reason for rejecting the paper.

Another issue is a lack more thorough discussion of how these findings compare with analogous studies in the reinforcement learning context.  It's a natural question to raise, and there isn't much commentary in the paper.

I think the paper leaves a lot of tantalizing questions unanswered, but that's as much the effect of clearly articulating the follow-up questions & speculations raised by the study, rather than anything else.  So I wouldn't hold that against the paper.

Minor stuff:

-- Typo on Line 242

-- Figure 2 is too small

**Additional Feedback:**

None.

**Documentation:**

I took a look at the github -- it seems reasonable.

**Ethics:**

No concerns.

**Relation To Prior Work:**

There isn't much direct prior work in this setting for imitation learning.  The closest area is reinforcement learning and there could be more discussion there.

**Summary And Contributions:**

This paper studies the effect of representation learning and data augmentation on imitation learning.  A range of methods were considered on both the representation learning and the data augmentation side, including both pre-training and joint training.  The biggest take-away from the paper is that data augmentatoin seems to be the single most impactful thing one can do to improve behavioral cloning.  This is a reasonable finding.

---

> ### Author Response · Authors · 2021-09-30
> **Response to reviewer 265a**
>
> Thank you very much for your helpful feedback and recognition of our work. We appreciate that you think our paper is valuable in (1) the provided framework with a wide range of design decisions, which can be a good template for future work, (2) the reasonable setup for a wide range of experiments, (3) the clarity of the paper, (4) the well-documented code, and (5) its reasonable conclusion. We address your comments and questions in the below.
>
> **Q1. “…how [well do] the results generalize to other imitation learning approaches beyond behavioral cloning.”**
>
> **A1.** This is a good point. We decided to add GAIL + RepL pretraining results to the paper as well, to complement the BC results. Refer to Table 6 in the main paper—note that three Procgen environments are missing because we could not get better-than-random performance out of our GAIL baseline, but we will continue tuning on these environments and include our best results in the camera-ready.
>
> The GAIL + RepL results in Table 6 are quite similar to the BC + RepL pretraining results in Table 4. Indeed, despite using a learned representation to initialize both the GAIL policy and the discriminator, we see even fewer significant deviations from the baseline than for BC. However, removing augmentations from GAIL does significantly decrease performance, which suggests that most of the value of a good representation in GAIL is obtained from augmentation alone. We suspect, but have not confirmed, that these results will also be consistent with those of similar “adversarial”-type IL approaches, such as Wasserstein GAIL and apprenticeship learning.
>
> **Q2. “…more thorough discussion of how these findings compare with analogous studies in the reinforcement learning context.”**
>
> **A2.** Thank you very much for your suggestion. We have added a section (Appendix B) summarizing key features of them and their major conclusions. We hope this can make it clearer on the differences of these works and their contributions. They basically differ on whether they break down RepL algorithms into different design components, the domain of concern (RL or IL), the number of environments and algorithms evaluated, and whether they experiment on image-based environments, and whether they explicitly compare with an image augmentation baseline.
>
> Thanks for pointing out typos and problems with figure sizes; we have now fixed these issues.

---

### Author Response · Authors · 2021-09-30
**New revision**

Thanks to all of the reviewers for their helpful comments. We’re glad that the reviewers appreciated the significance of our setting, the wide range of our experiments, and the quality and extensibility of our codebase. We have uploaded a new revision incorporating most of the improvements suggested by reviewers, which we summarise below.

The most commonly requested change was to run more experiments, whether with additional IL algorithms or additional environments. To address this, we have added results with another IL algorithm, GAIL. We found similar results for GAIL as for BC: augmentations often help substantially (and are sometimes essential to achieving better-than-random performance), but RepL almost never yields significant improvement in mean return. We also attempted to add CARLA results, which Reviewer zMjo suggested would be a useful addition. However, were not able to match previously reported results for BC performance during the rebuttal period—in particular, we are using the [D4RL](https://arxiv.org/pdf/2004.07219.pdf) CARLA tasks and dataset, but have only been able to achieve a mean return of 13 (without repL) or ~0 (with repL), compared to the D4RL paper’s reported mean return of 324 (for carla-lane). We suspect that this is just a hyperparameter issue, and will continue debugging it in the hope that we can include CARLA results in the camera-ready.

In addition, we made several other suggested improvements:

- Re-worded and sharpened the hypotheses in Section 4 in response to Reviewer bFea’s points.
- Added ablations for momentum, different projection heads, and compression, as requested by two reviewers.
- Fixed the typos and readability issues that were helpfully caught by the reviewers.

---

### Decision · Program_Chairs · 2021-10-09

**Decision:**

Accept

**Comment:**

This work provided a systematic investigation of the efficacy of representation learning for imitation learning. The experiment settings are well thought of and the benchmark is designed in a modular fashion with great extensibility for future work. The results have indicated that several recent advanced representation learning methods have led to marginal gains over the simple baseline of image augmentation when used with an imitation learning algorithm (behavioral cloning). While this is a bit of a negative result, such evaluations could potentially benefit the research community for validating the real merit of representation learning for policy learning. At the end of the discussion period, all four expert reviewers expressed positive impressions of this work. The AC agreed with the reviewers that this work could be further strengthened with 1) expanded discussions on the experimental results, 2) evaluations with other imitation learning algorithms, and 3) testing on more realistic decision-making domains (e.g., autonomous driving and robot manipulation). Nonetheless, these suggestions should not be considered as a ground for rejection. The AC believes that this kind of systematic empirical evaluation, albeit with negative results, should be encouraged in our community and thus recommends the acceptance of this paper.